# Position: AI Usage Policies Should Be Aligned with International Human Rights Law

**Jordi Calvet-Bademunt** [1,2]

## Abstract

Concerns about misinformation and disinformation are central to debates on the governance of generative AI services, yet guidance on when and how providers should restrict mis/disinformation while respecting freedom of expression remains underdeveloped. AI usage policies are a primary mechanism of user guidance and, in practice, operate as a form of private speech governance with direct implications for users' ability to seek, receive, and impart information. Building on international human rights law—especially ICCPR Article 19 and its legality, legitimacy, and necessity/proportionality requirements—this position paper proposes a set of checkable criteria for evaluating disinformation-related restrictions in usage policies, in a way that machine learning teams can operationalize when drafting rules and enforcement guidance. We apply the criteria to a comparative snapshot of eight leading providers' public policies (as of January 21, 2026) and find recurring shortcomings, including vague prohibitions, under-specified theories of harm, and limited articulation of less-restrictive alternatives. We argue that aligning usage policies with Article 19 can improve clarity and consistency, constrain overreach, and offer a principled basis for managing disinformation risks in AI-mediated information environments.

## 1. Introduction

Disinformation and misinformation have emerged as prominent concerns in the governance of generative AI systems (Bentzen, 2025). Large language models can enable the creation and dissemination of false or misleading information at scale, including through targeted and automated outputs, potentially amplifying downstream harms (Williams, 2024) (OpenAI, Georgetown University Center for Security and Emerging Technology, and Stanford Internet Observatory, 2023). Deepfakes provide a salient illustration of these risks, but the challenge extends across modalities, including text and voice (Bentzen, 2025, pp. 3–5).

Although no single, universally accepted definition exists, "misinformation" is commonly understood as false or inaccurate information shared without intent to deceive, whereas "disinformation" refers to false or inaccurate information deliberately spread to mislead (American Psychological Association, n.d.). This paper uses the term "mis/disinformation" to encompass both phenomena.

A wide range of governance and academic risk frameworks identify mis/disinformation as a salient risk associated with generative AI, including the OECD AI Principles (Organisation for Economic Co-operation and Development, n.d.), the European Union's AI Act (European Parliament and Council of the European Union, 2024), and the MIT AI Risk Repository (MIT AI Risk Initiative, 2025). Wachter et al. (2024) have further argued that AI systems should be subject to stronger epistemic duties, such as duties of truthfulness or non-deception. Yet despite frequent calls for AI companies to "limit misinformation," these frameworks provide limited guidance on when restrictions on expression are justified and how such restrictions should be designed in a manner consistent with freedom of expression and access to information.

This paper complements, rather than displaces, existing work in AI safety and alignment. Contemporary approaches—including responsible AI principles, risk-based governance models, and safety-by-design techniques such as red-teaming—have been instrumental in identifying and mitigating harms associated with machine learning systems. However, these approaches often prioritize precaution and harm prevention without articulating principled constraints on overreach, particularly in cases involving lawful but contested expression (Del Campo et al., 2025). International human rights law supplies these missing constraints by setting out cumulative requirements of legality, legitimacy, and

---

[1]The Future of Free Speech, Vanderbilt University, Nashville, Tennessee, United States [2]ESADE Law School, ESADE, Barcelona, Spain. Correspondence to: Jordi Calvet-Bademunt <jordi.calvet.bademunt@vanderbilt.edu>.

*Proceedings of the 43rd International Conference on Machine Learning*, Seoul, South Korea. PMLR 306, 2026. Copyright 2026 by the author(s).

necessity and proportionality for restrictions on expression (Calvet-Bademunt et al., 2025, pp. 16–17). Integrating this framework therefore provides a normative complement to existing safety approaches, helping distinguish legitimate risk mitigation from excessive or indeterminate restrictions in the governance of generative AI systems.

Concerns about mis/disinformation may be addressed at multiple stages of the generative AI lifecycle, including model design, deployment safeguards, distribution controls, and downstream platform policies. This paper focuses on usage policies—the publicly stated rules governing user interactions with generative AI models—because they are the primary public instrument through which providers communicate what uses are permitted or restricted (Klyman, 2024, p. 1). While broader frameworks exist for conducting human-rights impact assessments in AI systems (Business for Social Responsibility, 2025) (Committee on Artificial Intelligence (CAI), Council of Europe, 2024), they offer limited operational guidance on how to reconcile safety objectives with freedom of expression in concrete policy design. As generative AI systems increasingly mediate information access and inquiry (Anthropic, 2026) (OpenAI, 2025b) (Perez, 2025), usage policies play a growing role in shaping the practical boundaries of expression for large populations of users.

**This position paper proposes aligning usage policies with international human rights law and, in particular, Article 19 of the International Covenant on Civil and Political Rights (ICCPR)**, which protects the fundamental right to freedom of expression. This paper makes four contributions oriented toward both policy design and evaluation practice:

1. Normative specification: It proposes Article 19 of the ICCPR—a key international human rights instrument—as a globally applicable baseline for usage-policy design in speech-relevant AI contexts.

2. Operationalization: It translates the requirements of legality, legitimacy, and necessity into a benchmark with checkable criteria that can be used to analyze existing policies and guide clearer rule-drafting.

3. Case-study application: It applies the framework to mis/disinformation as a hard case and provides a comparative snapshot of eight major providers' public usage policies (as of January 21, 2026).

4. Research agenda: It identifies testable questions—including policy–behavior gap audits—for assessing overbreadth, transparency, and least-restrictive interventions in large language model deployments.

This paper does not claim that usage policies alone can resolve mis/disinformation, nor does it assess internal or unpublished company guidance. Instead, it evaluates usage policies as public governance commitments that shape user expectations and influence enforcement and model behavior. It further argues that divergences between publicly stated policies and model-level restrictions raise foreseeability and accountability concerns—core rationales of legality—and therefore warrant systematic evaluation in their own right.

Conflict of Interest Disclosure. The author participated in a research project — funded by Anthropic — that reviewed Anthropic's models on freedom of expression. The project received limited funding provided to the author's department, and the author received no personal remuneration. Anthropic had no role in the design, conduct, or conclusions of this paper, and the paper assesses Anthropic alongside seven other providers under the same publicly stated criteria.

## 2. Why Align AI Usage Policies with International Human Rights Law?

Generative AI systems are rapidly becoming central intermediaries for accessing, producing, and transforming information (Simon et al., 2025) (Anthropic, 2026). As model providers increasingly determine what users can and cannot generate, their usage policies function as a form of private speech governance with significant consequences for expression. These consequences are especially salient when users rely on AI systems for journalistic, academic, or civic inquiry, or to develop arguments on contested issues. Overbroad restrictions framed in terms of "mis/disinformation" risk unduly narrowing the range of lawful information and viewpoints users can explore, particularly in domains where uncertainty, disagreement, or evolving evidence is common (e.g., public health controversies, climate accountability narratives, historical disputes, or contested claims about war crimes and atrocities).

Many AI safety and risk frameworks already treat mis/disinformation as a salient risk in the generative AI ecosystem, including in emerging governance efforts for general-purpose models (European Commission, 2025). But these frameworks often leave underspecified a key design question: when restrictions on expression are justified and how to implement them in a way that avoids overreach, supports transparency, and provides predictable guidance to users.

International human rights law provides a principled framework for answering that question. Under Article 19 of the ICCPR, freedom of expression—including the freedom to seek, receive, and impart information regardless of frontiers and through any medium—is presumptively protected (United Nations General Assembly, 1976). Restrictions are permissible only if they satisfy cumulative requirements of legality, legitimacy, and necessity/proportionality (Human

Rights Committee, United Nations, 2011). This structure maps naturally onto usage-policy design: it offers a disciplined way to weigh expressive freedom against legitimate safety concerns, and to distinguish justified risk mitigation from vague, indeterminate, or excessive restrictions.

Three considerations make international human rights law a particularly appropriate baseline for usage-policy governance.

First, maturity and specificity. Article 19 is supported by decades of interpretive guidance clarifying the scope of protected expression and the narrow conditions under which restrictions are allowed (Columbia Global Freedom of Expression, n.d.). This makes it well-suited for translating broad, often elastic notions invoked in AI governance—such as "safety," "harm," or "misinformation"—into constrained, accountable rules. It also helps foreground well-documented risks of disproportionate or uneven enforcement in content governance, including for marginalized communities (Haimson et al., 2021), and specific languages (Dwoskin, 2022), and provides a vocabulary for evaluating foreseeability and consistency in policy application.

Second, global applicability. Leading AI providers operate across jurisdictions with divergent legal standards and political commitments. Anchoring usage policies in international human rights law helps avoid implicitly exporting the speech norms of any single state as a de facto global standard. Given its wide ratification (UN OHCHR, 2026), the ICCPR approximates a globally endorsed baseline for expressive freedom and a common language for cross-border policy design.

Third, resilience against illegitimate pressure. "Disinformation" rules have frequently been used to suppress lawful dissent, criticism, and investigative reporting (Lim & Bradshaw, 2023). Similar dynamics can arise in the generative AI context, including demands to refuse or restrict content that is politically inconvenient but lawful (Reuters, 2025) (Ray, 2024). A rights-based baseline provides providers with a principled reference point for evaluating and, where appropriate, resisting censorship demands that conflict with international standards, a point also made by one of the leading United Nations experts on freedom of expression (Khan, 2021).

These arguments also align with the broader expectation—reflected in the UN Guiding Principles on Business and Human Rights—that companies have the responsibility to respect human rights and address adverse impacts connected to their activities (Office of the United Nations High Commissioner for Human Rights, 2011). Several major technology companies already make public human-rights commitments or reference rights standards in their governance frameworks (Google, n.d.) (Meta, n.d.a). This paper builds on that trajectory by specifying how international human rights law can be operationalized in the concrete, user-facing domain of AI usage policies, where safety claims and expressive constraints most directly shape what users can do.

## 3. What It Means to Align Usage Policies with International Human Rights Law

Aligning AI usage policies with international human rights law entails translating the structure of Article 19 of the ICCPR into operational rules that govern how models respond to user prompts and how restrictions are articulated, justified, and applied. Rather than importing legal doctrine wholesale, this approach treats Article 19 as a design framework for speech-relevant policy choices. Table 1 summarizes how the framework is operationalized for usage-policy analysis.

Under Article 19, restrictions on expression are permissible only if they satisfy a cumulative three-part test:

1. Legality. Restrictions must be clear, accessible, and sufficiently precise to enable users to foresee what content is restricted and under what conditions (Human Rights Committee, United Nations, 2011, para. 22).

2. Legitimacy. Restrictions must pursue a recognized aim, such as protecting the rights or reputations of others, or safeguarding public order, public health, or public morals (and, for states, national security) (Human Rights Committee, United Nations, 2011, para. 27).

3. Necessity and proportionality. Restrictions must be the least speech-intrusive means capable of addressing the identified harm, and must be calibrated to its severity and likelihood (Human Rights Committee, United Nations, 2011, para. 34).

Although Article 19 formally governs restrictions "provided by law" by states, the underlying rationales of the legality requirement—clarity, accessibility, and foreseeability—apply with particular force to private rule systems that operate as de facto constraints on expression at scale. Usage policies fall squarely within this category: they structure what users may ask, receive, or generate through widely deployed AI systems and shape both enforcement practices and model behavior.

Viewed in this way, aligning usage policies with international human rights law does not imply that private companies assume the role of states. Rather, it provides a principled method for evaluating and designing expressive constraints in systems that mediate access to information for large populations. The framework helps distinguish justified safety-motivated restrictions from vague, indeterminate, or overly broad prohibitions, and supports more transparent, predictable, and accountable governance of AI outputs.

*Table 1.* IHRL framework and implications for AI usage policies.

| Element | IHRL standard | Implications for AI usage policies | Examples |
|---|---|---|---|
| Default rule | Expression is presumptively protected | Allow outputs unless they fall within clearly defined, publicly listed restrictions | Political commentary, satire, criticism, artistic expression |
| Restriction test | Legality + legitimacy + necessity / proportionality | Policies should mirror this structure and apply it consistently | Restrictions cannot be based on mere controversy |
| Legality | Clear, public, precise rules | No "secret rules"; defined categories; predictable enforcement | Avoid "harmful content" as a standalone category |
| Legitimacy | Only recognized aims justify restrictions | Each prohibited category should state the concrete aim or harm | Public health, fraud, discrimination |
| Necessity / proportionality | Least restrictive effective measure | Graduated interventions (contextualization, friction, labels/provenance) before bans | Labeling political deepfakes vs. categorical bans |

# 4. Case study: Applying International Human Rights Law to Mis/disinformation

## 4.1. Why Mis/disinformation Is a Hard Case

Mis/disinformation is a particularly instructive case study for assessing how international human rights law can guide AI usage-policy design. Unlike categories such as child sexual abuse material or direct incitement to violence, false or misleading claims are, in principle, protected expression under Article 19 of the ICCPR (Khan, 2021, para. 38). Restrictions therefore raise acute freedom-of-expression concerns and require careful justification under the cumulative requirements of legality, legitimacy, and necessity and proportionality.

Approaches to mis/disinformation vary sharply across political contexts. In many democracies, false or misleading statements are generally protected unless they are linked to concrete harms and satisfy strict conditions for restriction (Brannon, 2022) (Verza, 2022, pp. 11–12). In more authoritarian settings, vague notions of "misinformation" are frequently used to suppress lawful dissent, investigative reporting, or criticism of public authorities (Lim & Bradshaw, 2023). This divergence illustrates the risks of delegating broad and ill-defined discretion over "disinformation" to private actors operating at global scale. At the same time, an increasing number of democracies have adopted disinformation-related measures, and freedom-of-expression experts have repeatedly warned about abuse, overreach, and chilling effects (Ó Fathaigh et al., 2025, p. 3).

UN human rights bodies have issued guidance directly relevant to this debate. The Human Rights Committee has emphasized that international law does not permit general prohibitions on false information or erroneous interpretations of past events (Khan, 2021, para. 38) (Human Rights Committee, United Nations, 2011, para. 49). Protection extends not only to accurate or widely accepted information, but also to erroneous statements, controversial claims, satire, and interpretations that may shock, offend, or disturb (Khan, 2021, para. 38). Similarly, the UN has cautioned against disinformation rules that are vague or overbroad, and has emphasized that restrictions should be tied to clearly articulated harms and a legitimate aim under Article 19 (Khan, 2021, paras. 54–55).

At the same time, mis/disinformation can cause serious harms in specific contexts—public health, democratic processes, and the rights of others. For this reason, several policy bodies (including EU expert initiatives) have emphasized prioritizing non-restrictive responses—such as transparency and media literacy—over broad prohibitions (European Commission, 2018). This combination—presumptive protection, high contextual variance, and real but contingent harms—makes mis/disinformation a demanding test case for whether usage policies can be both rights-respecting and safety-relevant.

## 4.2. Translating Article 19 into Usage-Policy Design for Mis/disinformation

Operationalizing Article 19 for mis/disinformation is, in practice, a question of how systems intervene under conditions of epistemic uncertainty and political contestation. A key implication is that the mere fact that a claim is disputed, unpopular, or likely false is not sufficient to justify restriction. What matters is whether the output is plausibly linked to a legitimate aim under Article 19 (e.g., protection of the rights of others, public health, or public order), and whether the intervention is the least speech-intrusive effective response.

Three design implications follow.

First, "mis/disinformation" should not function as a catch-all category. Policies are more rights-consistent when they define mis/disinformation with sufficient specificity to support foreseeability—i.e., so that users can anticipate whether content is restricted (Khan, 2021, para. 70). As an illustration from adjacent social-media content moderation, Meta's Community Standards define misinformation as content

with a claim determined to be false by an authoritative third party (e.g., public health authorities) (Meta, n.d.c). This reference is not an endorsement of Meta's approach; it is used to show that more operational definitions are feasible in practice.

Second, restrictions should be tied to identifiable harms and domains (Khan, 2021, para. 40). Usage policies are more transparent and auditable when they specify why certain mis/disinformation is restricted (e.g., public-health protection, fraud) rather than treating falsity as inherently prohibited. Harm-linkage also clarifies which Article 19 aims are being invoked.

Third, necessity and proportionality favor non-restrictive interventions wherever effective (Human Rights Committee, United Nations, 2011, para. 34). Especially for general informational queries, contextualization, disclosure requirements, friction, or redirection may be less speech-intrusive than outright refusal. Refusals are more defensible in narrow, high-risk cases where deception and concrete harm are integral (e.g., impersonation, coordinated influence operations, medical deception, scams). Policies can reflect the relevance of intent without requiring mind-reading by targeting observable deceptive tactics and high-risk use-cases, while permitting good-faith inquiry and discussion of contested claims.

### 4.3. Do Current Usage Policies Meet International Human Rights Law Requirements?

To assess alignment with Article 19, we reviewed publicly available usage-policy documents from Alibaba (2025), Anthropic (2025c), DeepSeek (2025), Google (2024), Meta (n.d.b), Mistral AI (2025), OpenAI (2025c), and xAI (2025) as of January 21, 2026.

To make the assessment operational, we develop a human-rights benchmark and apply a falsifiable decision rule: we evaluate a provider under this benchmark only if its usage policy explicitly refers to "misinformation" and/or "disinformation." Where the term appears, we assess whether the policy (i) defines the term with operational specificity (legality), (ii) links restrictions to a recognized Article 19 aim (legitimacy), and (iii) articulates differentiated measures consistent with least-restrictive intervention (necessity/proportionality). We also record whether intent is treated as a scope-narrowing or risk-relevant factor, as intent can be relevant to proportionality when tied to observable deceptive conduct.

To make this scheme reproducible, we apply five coding rules. Explicit reference asks whether the policy invokes "misinformation" or "disinformation." Term defined asks whether the policy narrows the category through a definition or limiting criteria, rather than relying on examples or adja-

cent concepts. Harm-linkage asks whether each restriction is tied to a specific harm of the kind recognized under Article 19(3) (e.g., the rights of others, public health, public order). Necessity and proportionality asks whether the policy articulates a graduated intervention logic, rather than presenting all violations as triggering the same response. Intent asks whether the policy distinguishes deliberate deception from inadvertent error or contested factual claims. We code each criterion on a three-level scale—✓ (clearly present), △ (partial or implicit), × (absent)—with textual evidence drawn from the policy documents compiled in Appendix 1.

We acknowledge two limitations of this coding scheme. First, even with explicit criteria, applying them to policy text requires interpretive judgment. The harm-linkage criterion, for example, asks whether a restriction is tied to a recognized Article 19(3) aim, but providers may reasonably read general references to "deception" or "manipulation" as implicitly invoking such aims, while a stricter reading would require explicit articulation. We adopt the stricter reading because legality and legitimacy under Article 19 emphasize foreseeability and explicit justification, but we recognize that different coders may draw the line differently. To mitigate this, we (a) provide the underlying policy text in Appendix 1 so that each coding decision can be independently checked, (b) use a three-level scale (✓ / △ / ×) that distinguishes clearly present, partial or implicit, and absent signals rather than forcing a binary judgment, and (c) restrict assessment to policies that explicitly reference mis/disinformation, avoiding speculation about silent provisions. The explicit-reference rule we use also excludes policies that address adjacent concepts—such as DeepSeek's prohibition on "false information" or xAI's restrictions on misleading outputs—without using the terms "misinformation" or "disinformation." Extending the benchmark to such adjacent provisions is a useful direction for future work. Second, the benchmark evaluates public policy text, not the model-level enforcement that may sit behind it; this is a deliberate scope choice (see Section 4.4 and Section 6), but it means our findings speak to what providers publicly commit to, not to what their systems actually do. Both limitations are consistent with the framework's broader purpose, articulated in Section 5 (Objection 4): aligning usage policies with Article 19 disciplines discretion rather than eliminating it.

Five of the eight providers explicitly reference mis/disinformation in their usage policies: Alibaba, Anthropic, Google, Meta, and Mistral AI. Table 2 summarizes their alignment with the benchmark, while Appendix 1 compiles the relevant usage policies for all eight companies assessed. DeepSeek, OpenAI, and xAI do not explicitly reference mis/disinformation in their policies.

(1) Legality: definitional clarity is the exception, not the

*Table 2.* Mis/disinformation in usage policies: human rights operationalization benchmark

| Provider | (i)
Explicit reference to mis/disinformation | (ii)
Term sufficiently defined | (iii)
Restriction linked to a specific reason covered by Art. 19(3) | (iv)
Necessity / proportionality considered | (v)
Intent considered |
|---|---|---|---|---|---|
| Alibaba | ✓ | △ | × | × | △ |
| Anthropic | ✓ | △ | △ | × | △ |
| Google | ✓ | △ | ✓ | × | ✓ |
| Meta | ✓ | × | × | × | ✓ |
| Mistral AI | ✓ | △ | △ | × | ✓ |

*Legend:* ✓ = clearly present; △ = partial / implicit; × = absent.

rule

Across the five providers that invoke the category mis/disinformation, there are no detailed and operational definitions. Mistral AI and Alibaba stand out for defining misinformation in terms of deliberate or intentional falsity and misleading presentation. For instance, Mistral AI's policies state: "Misinformation. You shall not use the Mistral AI Products to generate content that is deliberately misleading, false, or intended to deceive others." Even here, greater specificity would improve foreseeability, particularly regarding how falsity or misleading presentation is determined and how intent is operationalized.

The need for clarity is especially salient where users may reasonably worry that mis/disinformation could be applied to restrict politically sensitive viewpoints. For example, statements such as "pandemic-related restrictions in schools did more harm than good" are not inherently false, but their accuracy depends on metrics and normative judgments—such as how to weigh children's educational interests against public-health protections for vulnerable populations—that are often contested. Whether such claims are treated as permissible generalizations, incomplete but accurate descriptions, or misleading presentations is rarely specified in advance. In the absence of clear criteria—such as evidentiary thresholds, requirements for contextualization, or standards for inferring intent—the boundary between lawful political argument and sanctionable mis/disinformation remains difficult for users to anticipate.

Anthropic and Google rely primarily on examples, adjacent concepts, or domain-specific illustrations rather than definitions. Anthropic, for instance, does not allow users to "[c]reate or disseminate conspiratorial narratives meant to target a specific group, individual or entity" (Anthropic, 2025c). Google prohibits, among other categories, "[f]acilitating misleading claims related to governmental or democratic processes or harmful health practices, in order to deceive" (Google, 2024).

Meta invokes "disinformation" without defining it. From

a human-rights perspective, undefined or loosely bounded categories increase the risk of unpredictable, viewpoint-sensitive, or jurisdiction-sensitive enforcement, especially in contested factual domains.

(2) Legitimacy: harm-linkage is often implicit rather than explicit

Article 19 permits restrictions only for recognized aims, such as protecting the rights of others, public health, or public order. Among the reviewed policies, Google most clearly links restrictions to concrete harms in sensitive domains (e.g., fraud, public health, and democratic processes) that can be mapped onto Article 19 aims.

Anthropic and Mistral AI provide legitimacy signals primarily in implicit form, for example by referencing deception or manipulation without consistently specifying the corresponding harm or the relevant Article 19 aim across all prohibited categories. For instance, Mistral AI prohibits the generation of false information about "health," which reflects a legitimate aim, but also about "current events." The latter is overly vague and risks unjustified restrictions on matters of public interest, such as governmental responses to a catastrophe or a protest. As a result, legitimacy may exist in substance but remains under-specified in form, limiting transparency and external auditability. By contrast, Alibaba and Meta do not clearly articulate the harms that their mis/disinformation provisions are intended to address, making them particularly problematic.

(3) Necessity and proportionality: the most consistent gap

Necessity and proportionality are the most weakly articulated elements across the reviewed policies. None sets out a structured least-restrictive response logic in usage-policy text, for example, specifying when to contextualize or label content, when to introduce friction or disclosure requirements, and when refusal is warranted.

We treat an articulated least-restrictive logic as present only where a policy (a) identifies more than one type of intervention (e.g., contextualization, labeling, friction, refusal), (b)

specifies the conditions under which each is selected, and (c) presents these interventions as a risk-calibrated sequence rather than a flat prohibition. No reviewed policy meets all three sub-criteria, which is why the column reads "✗" across the board.

Alibaba, Anthropic, and Google include partial signals, notably references to blocking or modifying outputs, imposing user disclosure requirements, or employing balancing language. Anthropic, for instance, indicates that multiple responses may follow a usage-policy violation, including throttling, suspending, or terminating access to its products, as well as blocking or modifying model outputs. Google clarifies that exceptions may be made for educational, documentary, scientific, or artistic purposes, or where potential harms are outweighed by substantial public benefit. These exceptions, however, are not presented as a coherent escalation framework keyed to risk. Companies do not indicate under which instances more or less restrictive responses may be applicable.

From an international human rights perspective, the absence matters for two reasons. Substantively, it raises overbreadth concerns where refusals may be used even when less restrictive measures would be effective. Procedurally, it also implicates legality: without knowing how interventions escalate, users cannot foresee how requests will be treated, and external observers cannot evaluate whether less speech-intrusive measures were considered.

Where mis/disinformation is explicitly regulated, intent ("deliberate," "intentionally misleading," "to deceive") is frequently invoked as a scope-narrowing factor. Intent can be a rights-consistent proxy when tied to observable conduct (e.g., fabricated provenance, impersonation, coordinated manipulation). However, without accompanying guidance on least-restrictive measures, intent risks becoming discretionary or unenforceable, undermining both predictability and accountability.

Several providers articulate proportionality-like commitments outside usage policies—in system cards, safety notes, or transparency materials—suggesting an emerging least-restrictive logic at the model-behavior level. For instance, Anthropic notes in the system card for Claude Sonnet 4 that testing on sensitive topics showed more nuanced and detailed engagement than earlier versions, with the model more often offering high-level information in response to ambiguous prompts rather than defaulting to refusal (Anthropic, 2025a, p. 15). The system card for Claude Sonnet 4.5 also assesses refusals in relation to sensitive topics (Anthropic, 2025b, p. 13). Google has likewise highlighted gains in helpfulness and instruction-following, explicitly aimed at reducing refusals of benign requests (Comanici et al., 2025, p. 21). Meta reports a marked reduction in refusals on contested political and social issues, falling from 7 percent in Llama 3.3 to under 2 percent in Llama 4 (Meta, 2025). OpenAI, for its part, has rolled out a revised safe-completion framework designed to curb unnecessary refusals (OpenAI, 2025a). Yet these commitments remain largely absent from usage-policy text, where clarity and accountability pressures are typically highest.

### 4.4. Do Companies without Mis/disinformation Policies Comply with International Human Rights Law?

DeepSeek, OpenAI, and xAI do not include prohibitions framed around the terms "misinformation" or "disinformation" in their usage policies, though each addresses adjacent concerns such as false information, fraud, impersonation, or misleading outputs. From an Article 19 perspective, the absence of an explicit mis/disinformation category is not inherently problematic. Given that mis/disinformation is, in principle, protected expression, refraining from categorical policy-based restrictions can be compatible with international human rights law.

Importantly, the absence of a mis/disinformation rule does not prevent providers from addressing genuine harms through narrower restrictions that are more straightforwardly justified—e.g., fraud, impersonation—and through restrictions related to incitement to discrimination or violence. From an international human rights law perspective, such harm-specific prohibitions are generally preferable to broad and ill-defined bans on mis/disinformation.

At the same time, the lack of explicit mis/disinformation language raises a distinct question: whether providers may nonetheless restrict content internally categorized as mis/disinformation without clearly informing users through publicly accessible rules. From a legality and transparency standpoint, undisclosed restrictions are concerning because they undermine foreseeability and accountability, which are core rationales of legality.

This concern can be especially salient where credible reporting suggests systematic viewpoint-based suppression at the model or system level. Where such restrictions exist but are not reflected in public policy, the gap itself becomes an object of evaluation: it may indicate that mis/disinformation operates as an implicit justification for content control without the transparency that Article 19's legality logic demands.

## 5. Alternative Views

Several objections may be raised to using international human rights law as a baseline for AI usage-policy design.

*Objection 1: AI companies are not legally bound by international human rights law and may adopt stricter standards for safety or reputational reasons.*

It is true that private companies are not parties to human

rights treaties in the same way states are. Yet in practice, purely discretionary moderation often produces costs on multiple fronts: it can generate backlash for both under- and over-enforcement (Vogels et al., 2020), reduce predictability for users if rules are not sufficiently detailed and clear, and increase vulnerability to political pressure, particularly where governments invoke "disinformation" as a pretext for censorship (Khan, 2021, para. 79). A rights-based baseline does not eliminate provider discretion, but it supplies clearer constraints, improves transparency and auditability, and offers a principled reference point for evaluating and resisting illegitimate demands. Moreover, companies such as Google and Meta have already committed to respecting international human rights (Google, n.d.) (Meta, n.d.a), and the UN Guiding Principles on Business and Human Rights instruct companies to avoid infringing human rights and to address adverse impacts connected to their activities (Office of the United Nations High Commissioner for Human Rights, 2011).

*Objection 2: Usage policies should track national law rather than global standards.*

A jurisdiction-by-jurisdiction approach is both normatively and practically problematic. Normatively, international human rights law rests on the premise that freedom of expression attaches to individuals by virtue of their humanity, rather than as a contingent grant by states (United Nations General Assembly, 1976). Practically, country-specific policy baselines would be operationally burdensome for global services and would fragment user expectations, particularly for cross-border systems accessed "regardless of frontiers." A single baseline grounded in international human rights law offers a more coherent and administrable approach, while still allowing providers to comply with localized legal obligations where required.

This does not preclude jurisdictional variation in implementation. Providers already localize parts of their governance: Google's political-content advertising policies, for instance, vary by region with jurisdiction-specific verification requirements and election silence periods (Google, 2026). The position advanced here is that international human rights law should serve as a floor beneath such variation. Localization above the floor may be legally required; what it should not do is displace the baseline protections that Article 19 establishes.

*Objection 3: AI-generated outputs fall outside freedom of expression because they are not produced by a human speaker.*

Even if the status of machine-generated speech is contested, international human rights protect not only the right to impart information but also the rights to seek and receive it (United Nations General Assembly, 1976, Art. 19). Usage policies and model-level constraints therefore impli-

cate users' rights when they limit access to information, arguments, or inquiry mediated through AI systems. The relevant rights claim is thus not that the model is a rights-bearing speaker, but that users' expressive and informational interests are affected by the rules governing what the system will provide.

Taken together, even if international human rights law is understood as a non-binding reference point rather than a source of legal obligation, it offers a uniquely well-specified and globally intelligible framework for structuring usage policies, evaluating overreach, and designing speech-relevant restrictions in a manner that is transparent, predictable, and amenable to empirical assessment.

*Objection 4: International human rights law is itself subject to interpretive pluralism, complicating its operational use.*

This objection is well-taken. Interpretive pluralism is a recognized feature of international human rights law more broadly (Donders, 2013), and Article 19 is no exception. Its interpretation has evolved through guidance documents, decisions, and special rapporteurs, and reasonable disagreement persists on questions such as the threshold of harm required for permissible restriction or the weight of local context. Our claim is not that international human rights law eliminates interpretive judgment, but that its legality–legitimacy–necessity/proportionality sequence narrows the range of defensible policy choices and requires providers to make their reasoning explicit and auditable. Pluralism within international human rights law is bounded by a structured cumulative test; pluralism within current usage-policy categories such as "harmful content" or "misinformation" is not. The framework's value lies in disciplining discretion, not eliminating it.

## 6. Research Agenda

Aligning usage policies with international human rights law yields testable evaluation questions directly relevant to machine learning practice. The framework supports three parallel empirical directions—each pairing a normative requirement under Article 19 with a measurable evaluation target.

(1) Policy–behavior gap audits. A first direction concerns whether model refusals and modifications correspond to publicly stated policy categories, supporting the legality requirement of foreseeability and accountability. Concrete metrics include (a) the proportion of refusals that map to a publicly stated policy category; (b) the proportion of prompts within each prompt class that receive the least-restrictive effective intervention before refusal; and (c) inter-rater agreement on the prompt taxonomy itself, as a measure of construct reliability. Where audits reveal divergence between policy text and model behavior, this divergence itself becomes an

object of evaluation: under Article 19's legality logic, restrictions that operate without public articulation undermine the foreseeability that the framework requires.

(2) Query-taxonomy benchmarks and the intervention ladder. A second direction operationalizes necessity and proportionality by systematically comparing model responses across distinct classes of prompts: (i) general informational queries, (ii) advocacy or argumentative generation, (iii) requests involving deceptive tactics or impersonation, and (iv) high-risk public-health or election-related prompts. A rights-consistent intervention ladder pairs each class with a graduated response: class (i) supports outputs by default; class (ii) supports outputs but may add contextualization where claims are contested; class (iii) escalates from disclosure to friction to refusal as deceptive intent or downstream harm becomes clearer; and class (iv) admits refusal as a defensible default where narrower interventions are unlikely to address the harm. This ladder operationalizes the necessity/proportionality requirement as a measurable design target: for any prompt class, evaluators can ask whether the system selected the least-restrictive effective intervention available to it.

(3) Cross-provider benchmarking of least-restrictive interventions. A third direction builds on the taxonomy to enable comparison across providers. Standardized prompt sets covering all four classes allow simultaneous assessment of overbreadth (refusals of general informational queries) and under-enforcement (failure to intervene on high-risk deceptive prompts). Concrete proportionality metrics include the ratio of contextualization to outright refusal across risk levels, and the rate at which intermediate interventions—such as disclosure, friction, or provenance cues—are deployed before escalation. Cross-provider benchmarking on shared prompt sets would allow the field to identify both convergent practice and outlier behavior, supporting more transparent comparisons than is currently possible from policy text alone.

These evaluation tasks have natural counterparts in model training and inference. Existing methodological paradigms already translate normative principles into computable structures: Constitutional AI (Bai et al., 2022) demonstrates how rule-based principles can be incorporated into training loops as constraints guiding model critique and revision, and extensions show that such principles can be derived from broader, collective inputs rather than from developer-defined preferences (Huang et al., 2024). Alignment with international human rights law can therefore be operationalized as a structured combination of (i) explicit normative constraints—drawn from Article 19's cumulative test—shaping optimization objectives or system prompts; (ii) procedural safeguards at inference time, such as escalation logic that maps prompt classes to graduated interventions; and (iii)

external governance and auditing mechanisms, including the policy–behavior gap audits proposed above.

This translation is non-trivial. Recent work documents the sociotechnical limitations of preference-based alignment in capturing plural and contested values (Dahlgren Lindström et al., 2025), the role of human and algorithmic discretion in interpreting normative principles (Buyl et al., 2025), and fundamental trade-offs in representativeness, robustness, and tractability across alignment approaches (Sahoo et al., 2025). The framework proposed here does not resolve these challenges, but it supplies a more disciplined target than category-level prohibitions and a clearer link between policy text and trainable objectives—connecting governance commitments, model behavior, and safety interventions in a way that supports systematic evaluation across providers.

## 7. Call to Action and Conclusion

Usage policies governing end-user interactions with generative AI systems increasingly function as a form of private speech governance. This paper has argued that such policies should be anchored in international human rights law—particularly Article 19 of the ICCPR—because it provides a coherent, globally applicable, and operational framework for balancing freedom of expression with legitimate safety concerns in AI-mediated information environments.

Implementing this framework requires a reorientation in policy design and evaluation. Expression should be permitted by default, with restrictions limited to narrowly defined, publicly articulated categories. Any constraint should satisfy cumulative requirements of legality (clarity and accessibility), legitimacy (pursuit of recognized aims), and necessity and proportionality (use of the least speech-intrusive effective intervention). The mis/disinformation case study illustrates the promise of this approach and the shortcomings of current practice, including vague prohibitions, under-specified theories of harm, and limited articulation of proportionality.

For the machine learning community, the contribution of this framework is not merely normative. Aligning usage policies with international human rights law enables testable evaluations of deployed systems—linking policy text, model behavior, and safety interventions—and supports more transparent comparisons across providers. We therefore call on model developers, evaluators, and governance researchers to treat usage policies as first-class design artifacts, to subject them to systematic audit and benchmarking, and to integrate rights-consistent constraints into safety and alignment workflows. Doing so can enhance transparency, reduce overreach, and strengthen the legitimacy and accountability of generative AI systems as they become central intermediaries of information and inquiry.

## Acknowledgements

The author thanks Jacob Mchangama and Isabelle Anzabi for their co-authorship on the prior work on which this paper builds, Axel Brando Guillaumes for his comments and guidance on this paper, and Rebeca Carpi Martín for her guidance on the broader research project of which this paper is part.

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

**Appendix 1**

Mis/Disinformation Provisions in AI Usage Policies

| Provider | Relevant policy excerpts |
|---|---|
| Alibaba | "1. Universal Usage Standards: Comply with the Applicable Laws: Do not use our services to engage in or facilitate illegal activities, including but not limited to: . . . Fraud and Misrepresentation: Engaging in fraudulent activities, impersonation, or spreading disinformation/misinformation."
"3. Prohibited Content and Activities: . . . 3.2 Misleading or Deceptive Practices: Disinformation: Do not generate or spread false information, conspiracy theories, or content intended to mislead or manipulate." |
| Anthropic | "Do Not Create or Spread Misinformation: This includes using our products or services to: create or disseminate deceptive or misleading information about, or with the intention of targeting, a group, entity or person; create or disseminate deceptive or misleading information about laws, regulations, procedures, practices, standards established by an institution, entity or governing body; create or disseminate conspiratorial narratives meant to target a specific group, individual or entity; impersonate real entities or create fake personas to falsely attribute content or mislead others about its origin without consent or legal right; provide false or misleading information related to medical, health or science issues." |
| DeepSeek | "If you publish or disseminate outputs . . . you must: (1) proactively verify the authenticity and accuracy of the output content to avoid spreading false information; (2) clearly indicate that the output content is generated by artificial intelligence." |
| Google | "Do not engage in misinformation, misrepresentation, or misleading activities. This includes: frauds, scams, or other deceptive actions; impersonating an individual (living or dead) without explicit disclosure in order to deceive; facilitating misleading claims of expertise or capability in sensitive areas (e.g., health, finance, government services, or law); facilitating misleading claims related to governmental or democratic processes or harmful health practices; misrepresenting the provenance of generated content by claiming it was created solely by a human, in order to deceive." |
| Meta | "Intentionally deceive or mislead others, including use of Llama 4 related to the following: generating, promoting, or furthering fraud or the creation or promotion of disinformation." |
| Mistral AI | "Misinformation. You shall not use the Mistral AI Products to generate content that is deliberately misleading, false, or intended to deceive others. This includes, for instance: spreading false information about current events, health, or science; undermining the integrity of a civic or political process; promoting conspiracy theories that can cause harm; misinformation that targets protected groups." |
| OpenAI | "Our services may not be used to manipulate or deceive people, including through deceit, fraud, scams, impersonation, or election interference and demobilization activities." |
| xAI | "Do not mislead people as to the nature and source of outputs. You should be transparent and disclose the use of AI assistance and potential limitations." |

