# OpenReview forum: "Position: AI Usage Policies Should Be Aligned with International Human Rights Law"
_ICML.cc/2026/Position_Paper_Track — ICML 2026 Position Paper Track regular_

### Official Review · Reviewer_UXay · 2026-03-11

**Significance:** 3
**Argument Clarity:** 3
**Rating:** 5
**Confidence:** 2

**Questions:**

- In objection 2 (sec. 5), the authors argue against jurisdiction-by-jurisdiction approach. Are the authors aware of generative AI providers, that adapt their policies or system behaviour depending on user location or national requirements?

**Alternative Views Section:**

Yes

**Compliance With Llm Reviewing Policy A Conservative:**

Affirmed.

**Discussion Potential:**

3

**Final Justification:**

The clarifications adequately addressed my remaining concerns regarding pluralism and meaningfully extension of the discussion in Section 6 as outlined in the rebuttal.

**Paper Summary:**

The papers’ advocated position is that generative AI systems usage policies should be aligned with international human rights law. To this end, the authors focus mainly on Article 19 of the International Covenant on Civil and Political Rights (ICCPR), which protects freedom of expression and defines conditions under which restrictions are justified, and the case of mis/disinformation. The paper discusses the rationale of aligning AI usage policies with international human rights law and implications. This is then illustrated on a case study on mis/disinformation and usage policies of various AI providers. Based on these observations, the authors propose a research agenda for evaluating the alignment between publicly stated usage policies and the actual behavior and usage restrictions of deployed models.

**Position:**

Yes

**Position In Title:**

Yes

**Related Work:**

3

**Strengths And Weaknesses:**

**Strength**
- In general, the position is clear and well-motivated, the paper is well structured and easy to follow.
- Translating the Article 19 principles into concrete criteria and the case study as a concrete example is helpful and makes the argument actionable
- Also, alternative views are well described and appropriately addressed.

**Weaknesses**
- The paper argues that international human rights law provides a common baseline, but in practice pluralism also exists within international human rights interpretation itself (for instance see https://doi.org/10.1093/acprof:oso/9780199642120.003.0008), which could complicate its operationalization in AI systems.

- Sec. 6 (Research Agenda) is relatively brief compared to the earlier sections. Since the paper is submitted to a machine learning venue, this section could more strongly discuss what concrete technical research directions follow from the proposed framework. For instance, the mentioned pluralism (independent of international or national law) introduces an implementation challenge: AI systems might need to adapt behavior depending on jurisdiction, user context, or legal requirements. That said, expanding the discussion in the research agenda could strengthen the contribution.

**Support:**

3

---

> ### Author Rebuttal · Authors · 2026-03-28
>
> We thank the reviewer for the supportive reading and for highlighting both the paper’s strengths and the remaining implementation questions. We address the two points and the question below.
>
> 1. On pluralism within international human rights interpretation
> This is a fair and important point. We do not deny that international human rights law leaves room for interpretive disagreement; the reference is well taken. Our claim is narrower: Article 19 provides a more structured and constrained baseline than the broad, often undefined categories currently used in AI usage policies. Even where pluralism remains, the legality–legitimacy–necessity/proportionality sequence narrows the range of acceptable policy choices and requires providers to articulate their reasoning in a more transparent way.
>
> 2. On the research agenda being brief
> We agree that this section can be made more concrete. The paper already identifies three technical directions—policy–behavior gap audits, prompt-taxonomy benchmarks, and evaluation of least-restrictive interventions—and these can be developed into specific ML research tasks. For example: (i) designing prompt-taxonomy datasets that instantiate the four prompt classes discussed in the paper, enabling standardized cross-provider comparisons; (ii) developing proportionality metrics—such as the ratio of contextualization responses to outright refusals across risk levels—to measure whether systems implement graduated interventions; (iii) cross-provider benchmarking on these standardized prompt sets to assess both overbreadth (refusals of general informational queries) and under-enforcement (failure to intervene on high-risk deceptive prompts); and (iv) the implementation challenge the reviewer identifies: how systems might adapt behavior depending on jurisdiction or user context while maintaining a human-rights baseline, which raises broader implementation questions about personalization, localization, and baseline consistency in safety systems. We will make this operational logic explicit in the camera-ready.
>
> Q: Are the authors aware of providers that adapt policies or behavior depending on user location or national requirements?
> While usage policies are typically global, other subject-specific policies and behaviour can change across regions. For instance, Google’s public political-content policies explicitly vary by region, including jurisdiction-specific verification requirements and election silence periods (https://support.google.com/adspolicy/answer/6014595?hl=en). Our point is not that localization never occurs, but that a human-rights baseline should serve as a floor beneath geographically differentiated implementation: localized compliance above that floor is expected, but localization should not displace the baseline protections that Article 19 provides.

---

> > ### Author Rebuttal · Reviewer_UXay · 2026-04-03
> >
> > Thank you for the rebuttal. The clarifications addressed my concerns and incorporating them into the paper will strengthen it further.

---

### Official Review · Reviewer_qofp · 2026-03-12

**Significance:** 3
**Argument Clarity:** 3
**Rating:** 4
**Confidence:** 3

**Questions:**

NA

**Alternative Views Section:**

Yes

**Compliance With Llm Reviewing Policy A Conservative:**

Affirmed.

**Discussion Potential:**

3

**Paper Summary:**

The authors examine a notable theme in AI governance: how usage policies for generative AI systems should regulate misinformation while respecting freedom of expression. The authors study an important concept—the alignment of AI usage policies with international human rights law, specifically Article 19 of the International Covenant on Civil and Political Rights (ICCPR). The paper proposes an operational framework translating the legality, legitimacy, and necessity/proportionality principles of Article 19 into concrete criteria for drafting and evaluating AI usage policies. The authors analyze the public policies of eight major AI providers and identify common deficiencies such as vague definitions of misinformation, weak articulation of harms, and the lack of proportional intervention strategies. The paper argues that adopting a human-rights–aligned framework can improve transparency, consistency, and accountability in AI governance.

**Position:**

Yes

**Position In Title:**

Yes

**Related Work:**

3

**Strengths And Weaknesses:**

Strengths

* The paper provides a structured operationalization of international human rights principles (legality, legitimacy, necessity/proportionality) and translates them into evaluable criteria for AI usage policy design.
* By focusing on generative AI governance and misinformation risks, the work addresses a pressing issue as AI systems increasingly mediate information access and discourse.
* The review of policies from multiple major AI providers offers concrete evidence that existing policies lack clarity and proportionality mechanisms, strengthening the paper’s practical implications.

Weaknesses
* The evaluation of provider policies is largely qualitative and descriptive; the paper does not provide a systematic scoring methodology or reproducible evaluation protocol.
* The study focuses only on publicly available policy text rather than actual model behavior or enforcement practices, leaving a gap between policy and system implementation.
* While relevant for governance discussions, the work offers limited direct contributions to machine learning methodology or system design, which may reduce its impact for technical ML audiences (e.g., ICML).

**Support:**

3

---

> ### Author Rebuttal · Authors · 2026-03-28
>
> We thank the reviewer for the thoughtful review and for recognizing the paper’s relevance. We address the three weaknesses below.
>
> 1. On the evaluation being qualitative and descriptive
> The case study is qualitative, but it is structured: the paper applies the same benchmark across providers, states the decision rule (evaluate only where the provider explicitly references mis/disinformation), uses a three-level coding scale (✓ / △ / ×), and provides the underlying policy text in Appendix 1. The applied benchmark is outlined in Table 2: for each provider, we assess five criteria derived from Article 19 (explicit reference, definitional clarity, harm-linkage, necessity/proportionality, intent). As explained in point (1.) of the rebuttal to C4H1, we will spell out these criteria in the camera-ready to make the scheme reproducible.
>
> 2. On the gap between policy text and actual model behavior
> We agree completely that this gap matters. Indeed, one of the paper’s central claims is that the gap itself is normatively and empirically important: if restrictions are applied in ways not reflected in public policy, users lose clarity and foreseeability, which are core rationales of legality under Article 19. That is why Section 6 proposes policy–behavior gap audits, prompt-taxonomy benchmarks, and evaluation of least-restrictive interventions as direct next steps for ML research. These are the concrete bridges from the normative framework to empirical ML practice.
>
> 3. On limited direct contributions to ML methodology
> We appreciate this concern. Our contribution is not a new method or algorithm, but a structured evaluation framework and research agenda for a governance problem that directly shapes deployed model behavior. Concretely, the paper provides: (i) a benchmark with checkable criteria for evaluating usage policies (Table 2), which can be applied by any research team to any provider; (ii) a prompt taxonomy (four classes ranging from general informational queries to high-risk deceptive-tactic prompts) that can serve as the basis for evaluation datasets; (iii) an intervention hierarchy ladder that will be included in the camera-ready (allow → contextualize → friction → refuse) that provides measurable targets for refusal-rate analysis; for instance, measuring the proportion of class-ii prompts (contested-issue advocacy) that receive contextualization versus outright refusal; and (iv) three specific empirical directions: policy–behavior gap audits (do refusals correspond to stated policy?), cross-provider benchmarking on standardized prompt sets, and evaluation of whether systems implement least-restrictive interventions before escalating. These are tractable evaluation tasks using standard ML methods, and they address a challenge—usage policies and safety-related refusals—that directly shapes model behavior for large user populations. We offer this as a concrete evaluation framework and research agenda for an ML-relevant governance problem that directly shapes deployed model behavior.

---

> > ### Author Rebuttal · Reviewer_qofp · 2026-04-05
> >
> > NA

---

### Official Review · Reviewer_C4H1 · 2026-03-13

**Significance:** 3
**Argument Clarity:** 3
**Rating:** 4
**Confidence:** 3

**Questions:**

1. Why is Article 19 a better baseline than other candidate frameworks for private AI governance, rather than simply one useful viewpoint among several candidates?

2. How should the framework handle cases where public usage policies are broad, but model behavior is more nuanced in practice?

**Alternative Views Section:**

Yes

**Compliance With Llm Reviewing Policy A Conservative:**

Affirmed.

**Discussion Potential:**

3

**Final Justification:**

My concerns have been well-addressed.

**Paper Summary:**

This paper proposes that the usage policies regarding misinformation and disinformation of AI providers should be aligned with international human rights law, particularly Article 19 of the ICCPR. Specifically, policies should not be just product terms; instead, they should be governance that can directly help users to seek and receive important information. The authors have summarized their contributions into four aspects: (1) the paper points out that Article 19 provides a globally principled basis for speech-relevant AI policy design, (2) it converts the Article 19 into evaluation criteria for usage policies, (3) it applies the framework to compare 8 major AI providers’ policies, and (4) it formulates a research agenda for policy-behavior gap audits. Overall, this paper identifies that existing policies often have vague definitions and harm linkage, therefore, a rights-based framework could enhance AI usage-policy design by improving clarity, transparency, predictability, and accountability.

**Position:**

Yes

**Position In Title:**

Yes

**Related Work:**

3

**Strengths And Weaknesses:**

# Strengths
1. This paper identifies a real gap between broad calls to “combat misinformation” and the lack of principled guidance on how to do so without overreaching. Nowadays, as the popularity of generative AI, the governance of model outputs through usage policies is an important question.

2. The paper’s framing that treats usage policies as a form of private speech governance is compelling, and it is also persuasive that human rights law offers constraints missing from many AI safety frameworks. The paper has done really well in explaining why legality, legitimacy, and necessity matter.

3. The research agenda section provides a nice connection to the machine learning community. This section provides some practical relevance beyond the theory.

4. The presentation of this paper is another strength as it is easy to follow.

# Weaknesses
1. Despite the compelling claims, the case studies are relatively shallow. Although the paper claims to apply a benchmark to eight providers, the analysis is mostly qualitative and descriptive. The results would be much stronger if the paper included a more systematic scheme with clear criteria and evidence. As written, the comparative evaluation reads more like an illustration than a rigorous analysis.

2. Despite the paper repeatedly claiming the framework to be concrete and operationalizable, the benchmark designed is still high-level. For example, it would be better if the authors could elaborate on what textual evidence would count as an articulated least-restrictive logic.

3. The paper’s evidence base is limited to public usage policies, which weakens the paper’s practical conclusions. Public policy texts are only one layer of governance, and some of the most important rights impacts may arise from hidden classifier rules, safety tuning, or product-specific refusal behavior. The paper recognizes this, but the scope limitation should be more prominently framed.

**Support:**

3

---

> ### Author Rebuttal · Authors · 2026-03-28
>
> We thank the reviewer for the balanced and thoughtful review. We are grateful for the recognition of the paper’s framing and its relevance to the ML community. We address the three weaknesses and two questions below.
>
> 1. On the depth of the case studies
> Table 2 already applies a structured scheme: for each provider, we assess five criteria derived from Article 19 (explicit reference, definitional clarity, harm-linkage, necessity/proportionality, intent), using a three-level coding scale and textual evidence drawn from the policy documents and Appendix 1. To make the coding logic explicit: “explicit reference” asks whether the policy invokes misinformation/disinformation or a close equivalent; “term defined” asks whether the policy narrows the category through a definition or limiting criteria; “harm-linkage” asks whether the restriction is tied to a specific Article 19(3)-type harm; “necessity/proportionality” asks whether the policy articulates a graduated intervention logic; and “intent” asks whether it distinguishes deception from mistake. We will spell out these coding rules in the camera-ready to make the scheme reproducible.
>
> 2. On the benchmark remaining high-level
> The paper’s unit of analysis is public policy text, so the benchmark asks what a provider publicly commits to, not what hidden enforcement systems may do. For the reviewer’s specific example, we would treat “articulated least-restrictive logic” as present only where a policy: (a) identifies more than one intervention type (for example, contextualization, labeling, friction, refusal); (b) specifies conditions for choosing among them; and (c) presents them as a risk-calibrated sequence rather than a flat prohibition. We will clarify this in the camera-ready version. No reviewed policy currently meets all three sub-criteria, which is why the column reads “×” across the board.
>
> 3. On the scope limitation to public policy text
> This is a genuine limitation, and we appreciate the suggestion to foreground it more clearly. Our choice is principled: public usage policies are the primary governance commitments visible to users, and they are where clarity, accessibility, and foreseeability matter most. The paper’s argument is not that policy text exhausts AI governance, but that it is a necessary and publicly auditable layer of it. Hidden classifier rules or safety tuning may matter greatly; the paper’s position is that divergences between those hidden layers and the public policy should themselves become an object of evaluation under the legality principle. Section 6’s research agenda calls for policy–behavior gap audits as the direct next empirical step.
>
> Q1: Why Article 19 rather than another framework?
> We do not argue that Article 19 is the only useful lens. Our narrower claim is that it has three comparative advantages for this specific design task: (i) maturity and specificity: decades of interpretive guidance (General Comment No. 34, Special Rapporteur reports) that alternatives like the OECD AI Principles, the EU AI Act’s risk categories, or academic risk taxonomies do not yet provide for speech-related policy design; (ii) global applicability: as a widely ratified international instrument, the ICCPR avoids exporting any single jurisdiction’s speech norms as a de facto global standard; and (iii) structural completeness: its cumulative three-part test constrains both under- and over-enforcement, whereas most AI governance frameworks focus on harm prevention without articulating principled constraints on overreach. In that sense, Article 19 complements rather than displaces broader AI governance frameworks.
>
> Q2: How should the framework handle broad policies with more nuanced model behavior in practice?
> This is an important practical question. If a provider’s policy broadly prohibits “misinformation” but its model actually handles contested claims with nuance (e.g., contextualizing rather than refusing), the framework would identify two distinct issues: the policy fails the legality criterion (users cannot foresee the actual treatment from the policy text), while the model behavior may in fact be closer to the necessity/proportionality standard. In other words, nuanced behavior is welcome, but it should be reflected in public policy so that users can anticipate how their requests will be treated and external observers can audit consistency. The policy–behavior gap audit proposed in Section 6 is designed to surface these mismatches.

---

> > ### Author Rebuttal · Reviewer_C4H1 · 2026-04-02
> >
> > My concerns have been well-addressed.

---

### Official Review · Reviewer_hDmZ · 2026-03-17

**Significance:** 3
**Argument Clarity:** 2
**Rating:** 3
**Confidence:** 3

**Questions:**

See weaknesses.

**Alternative Views Section:**

Yes

**Compliance With Llm Reviewing Policy A Conservative:**

Affirmed.

**Discussion Potential:**

3

**Final Justification:**

The position is clear and well justified. It would be strengthened to better translate the norms/standards into ML languages/workflows.

**Paper Summary:**

This paper proposes a framework with a set of checkable criteria for evaluating disinformation-related restrictions in generative AI usage policies, based on international human rights law. The authors reviewed usage-policy documents from eight major AI providers against this framework and conclude that current policies are vague. Finally, they propose a research agenda for the ML community to audit policy-behavior gaps and benchmark models against the taxonomies.

**Position:**

Yes

**Position In Title:**

Yes

**Related Work:**

3

**Strengths And Weaknesses:**

Strengths:
1. The paper focuses on addressing the black box of AI usage policies, which is a significant and timely topic for the AI community.
2. The paper provides a helpful translation of Article 19 into a checklist for qualitatively evaluating AI usage policies, and reviews existing policy documents from model providers to highlight the gap of how the industry currently handles misinformation.

Weaknesses:
1. While the authors argue for clearly-specified definitions, I did not see clearly how high-level IHRL standard is defined in AI policy languages without a concrete framework. The paper assumes that the human rights law is less vague than corporate policy, but legal terms are notoriously context-dependent and may also trigger debates. It can be strengthened if the paper shows how the law is turned into common standards that guide engineering workflows, otherwise it is still abstract for a machine learning audience/teams to operationalize.
2. The proposed research agenda suggests policy-behavior gap audits but lacks the rigor expected at ICML. For example, how is the audit conducted and what are the formal/actionable metrics for proposed requirements?


Suggested Improvements:
1. Move beyond high-level categories and provide a draft of a rights-consistent taxonomy for a specific hard case to demonstrate exactly how operational specificity looks in practice.
2. To make the position more achievable, the authors are encouraged to specify how this alignment with human rights law reshape model behavior with more specified standards. For instance, could/how this be framed as a defined constraints in model training/inference?

**Support:**

3

---

> ### Author Rebuttal · Authors · 2026-03-28
>
> We thank the reviewer for the careful engagement and constructive suggestions. We address the four central concerns below.
>
> 1. On operationalizing IHRL for ML teams
> We agree that the key question is not whether Article 19 is normatively attractive in the abstract, but whether it can guide concrete policy and system design. Our claim is not that international human rights law eliminates judgment, but rather that it supplies a more disciplined structure than current provider terms such as “misinformation” or “harmful content,” which are often undefined or vaguely defined and weakly tied to a cumulative justification logic. The paper operationalizes this logic in Table 1 and applies it in Table 2 through checkable criteria tied to legality, legitimacy, and necessity/proportionality.
> To make the operational content more explicit for an ML audience, a rights-consistent policy for the paper’s hard case—mis/disinformation—can distinguish four prompt classes already identified in Section 6: (i) general informational queries; (ii) advocacy or argumentative generation on contested issues; (iii) deceptive tactics or impersonation; and (iv) high-risk public-health or election-related prompts. A rights-consistent intervention ladder would then be: allow for (i); generally allow or contextualize for (ii); use disclosure/friction/refusal depending on risk for (iii); and apply the strongest intervention—including refusal when narrower tools are inadequate—for (iv). In engineering terms, these become policy labels for prompt categories, an intervention hierarchy, and evaluation targets for refusal vs. contextualization behavior. We would include this intervention ladder in Section 6.
>
> 2. On rigor and metrics for policy–behavior audits
> We agree that the research agenda should be as concrete as possible. The paper identifies three empirical directions—policy–behavior gap audits, query-taxonomy benchmarks, and evaluation of least-restrictive interventions—and these can be instantiated with standard ML-style metrics. For example, a policy–behavior audit could measure: (a) the proportion of refusals that map to a publicly stated policy category; (b) the proportion of prompts within each prompt class that receive the least restrictive effective intervention before refusal; and (c) inter-rater agreement on the prompt taxonomy. Our aim in Section 6 is to show that the framework yields concrete, testable evaluation questions for ML research, not to present a completed empirical study. We nevertheless agree that making those evaluation targets more explicit strengthens the paper, and we will do so in the camera-ready version.
>
> 3. On the draft taxonomy for a hard case
> The paper sketches these four prompt classes and a corresponding intervention logic; the clarification above makes that operational structure explicit. More generally, our position is that alignment with human rights law should reshape model behavior by replacing broad category-level bans with narrower prompt classes, explicit harm-linkage, and graduated interventions before refusal, providing both policy drafters and ML evaluation teams with clearer and more principled design targets than the current status quo.
>
> 4. Framing as a defined constraint in model training/inference
> We agree that the manuscript should more explicitly articulate how alignment with human rights law can be operationalized in model training and inference. In the revised version, we will add a paragraph clarifying that such alignment can be grounded in existing methodological paradigms that translate normative principles into computable structures. In particular, prior work on Constitutional AI demonstrates how explicit rule-based principles can be incorporated into training loops as constraints guiding model critique and revision (Bai et al., 2022, arXiv:2212.08073), while more recent extensions show that these principles can be derived from broader, collective inputs rather than developer-defined preferences (e.g., collective constitutional frameworks, arXiv:2406.07814). At the same time, we will emphasize that this translation is inherently non-trivial: recent literature highlights both the sociotechnical limitations of preference-based alignment methods such as RLHF in capturing plural and contested values (e.g., PubMed:40486676), and the role of human and algorithmic discretion when interpreting normative principles in practice (arXiv:2502.10441). Accordingly, we will frame alignment with human rights law not as an abstract desideratum, but as a structured combination of (i) explicit normative constraints or constitutions shaping optimization objectives, (ii) procedural safeguards at inference time, and (iii) external governance and auditing mechanisms to ensure consistent interpretation and enforcement, while acknowledging fundamental trade-offs in representativeness, robustness, and tractability identified in recent work on alignment complexity (arXiv:2511.19504).

---

> > ### Author Rebuttal · Reviewer_hDmZ · 2026-04-06
> >
> > Thanks for the detailed response. I increased my original score.

---

### Decision · Program_Chairs · 2026-04-30

**Decision:**

Accept (regular)

**Comment:**

While the reviewers are mostly positive about the paper, I am still not convinced about the "set of concrete, checkable criteria" that the authors posit; I still feel that Table 2 is not fully able to demonstrate this message. For instance, "Restriction linked to a specific reason covered by Article(13)" -- the platforms might still have their own interpretation of what is a "Restriction linked to a specific reason". It would be important that the authors acknowledge their limitations in the paper.